# Trends in Antibody Titers after SARS-CoV-2 Vaccination—Insights from Self-Paid Tests at a General Internal Medicine Clinic

**DOI:** 10.3390/medicines10040027

**Published:** 2023-04-20

**Authors:** Hiroshi Kusunoki, Kazumi Ekawa, Masakazu Ekawa, Nozomi Kato, Keita Yamasaki, Masaharu Motone, Hideo Shimizu

**Affiliations:** 1Department of Internal Medicine, Osaka Dental University, 8-1 Kuzuhahanazonocho, Hirakata 573-1121, Osaka, Japan; 2Department of Environmental and Preventive Medicine, Hyogo Medical University, Nishinomiya 663-8501, Hyogo, Japan; 3Ekawa Medical Clinic, Arida 649-0311, Wakayama, Japan; 4Department of Health and Sport Sciences, Graduate School of Medicine, Osaka University, Toyonaka 560-0043, Osaka, Japan; 5Faculty of Health Sciences, Osaka Dental University, Hirakata 573-1144, Osaka, Japan

**Keywords:** COVID-19, SARS-CoV-2, vaccines, breakthrough infection, SARS-CoV-2 antibody

## Abstract

**Background**: The rise in antibody titers against the novel coronavirus (SARS-CoV-2) and its duration are considered an important indicator for confirming the effect of a COVID-19 vaccine, and self-paid tests of antibody titer are conducted in many facilities nationwide. **Methods**: The relationship between the number of days after the second and third dose of vaccines, age, and antibody titer was determined from the medical records of general internal medicine clinics that conducted self-paid testing of the SARS-CoV-2 antibody titer using Elecsys Anti-SARS-CoV-2 S (Roche Diagnostics); the relationship between the number of days after two or more doses of vaccines and antibody titer was also determined. We also examined the antibody titers in cases of spontaneous infection with SARS-CoV-2 after two or more doses of the vaccine. **Results**: Log-transformed SARS-CoV-2 antibody titers measured within 1 month from the second or third dose of vaccine showed a negative correlation with age (*p* < 0.05). In addition, the log-transformed antibody titers also showed a negative correlation trend with the number of days after the second dose of vaccine (*p* = 0.055); however, there were no significant correlations between the log-transformed antibody titers and the number of days after the third dose of vaccine. The median antibody titer after the third vaccination was 18,300 U/mL, more than 10 times the median antibody titer after the second dose of vaccine, of 1185 U/mL. There were also some cases of infection after the third or fourth dose of vaccine, with antibody titers in the tens of thousands of U/ml after infection, but the patients still received further booster vaccinations after the infection. **Conclusions**: The antibody titers after the third vaccination did not attenuate after a short follow-up period of one month, while they tended to attenuate after the second vaccination. It is considered that many people in Japan received further booster vaccinations after spontaneous infection, even though they already had antibody titers in the tens of thousands of U/mL due to “hybrid immunity” after spontaneous infection following two or more doses of vaccine. The clinical significance of the booster vaccination in this population still needs to be thoroughly investigated and should be prioritized for those with low SARS-CoV-2 antibody titers.

## 1. Introduction

During the COVID-19 pandemic, the majority of the world’s population was infected with SARS-CoV-2 at least once. Nevertheless, the fatality rate decreased significantly since the start of the endemic phase [1]. The COVID-19 vaccine has been available in Japan since February 2021 for healthcare workers and since April 2021 for elderly citizens [2]. As of December 10, 2021, 77.3% of the total population had received two vaccination doses [3]. The third dose of vaccines also began to be administered in December 2021 for healthcare workers, and in early 2022, vaccinations began in earnest for the general population, starting with the elderly individuals.

Furthermore, from the end of May 2022, the fourth dose of vaccine was given after a gap of at least 5 months after the third dose to those aged 60 years or older, those aged 18 years or older with underlying medical conditions, and other patients diagnosed by physicians to be at high risk of severe COVID-19. 

Moreover, the vaccination interval for the BNT162b2 (BioNTech and Pfizer) and mRNA-1273 (Moderna and Takeda) vaccines was shortened from 5 months to 3 months starting in October 2022. Accordingly, the fifth dose of vaccine started. Thus, it can be said that COVID-19 vaccination has been promoted by government’s initiative in Japan.

Severe acute respiratory syndrome coronavirus 2 (SARS-CoV-2) antibody titers level and duration of elevated levels are considered important indicators of the effect of COVID-19 vaccines. Initially, the rise in antibody titers and its persistence were considered indicators not of a vaccine’s clinical effectiveness, such as in reducing the disease incidence and pre-venting its severe forms, but rather of its immunogenicity and biological post-immunization effects. Nevertheless, the current effect of the COVID-19 vaccination is usually evaluated by the rise in antibody titers. So, many facilities in Japan conduct self-paid tests for SARS-CoV-2 antibody titers. 

“Hybrid immunity” acquired by vaccination plus spontaneous infection brought about by breakthrough infection after vaccination is gaining worldwide attention as COVID-19 vaccination becomes more widespread [4]. A study by the Bahrain National Health Database showed that a “hybrid immunization” promoted by both vaccination-induced immunity and prior infections effectively reduced the severity of reinfection [5]. Similarly, a Swedish study showed that a “hybrid immunization” with one and two vaccinations reduced the risk of reinfection by 58% and 66%, respectively, compared to infection-induced immunization [6]. 

The following mechanisms were proposed to explain why “hybrid immunity” is effective in protecting against reinfection and in preventing its severe forms. For example, it has been reported that breakthrough infections after vaccinations significantly increased antibody titers [7,8,9] and that post-vaccination infections induced serum binding and neutralizing antibody responses that were significantly stronger, more prolonged, and more effective against the spike protein mutations than those observed with two-dose vaccination. In other words, compared to those vaccinated with only two vaccine doses, post-vaccination breakthrough-infected individuals have a robust serum-bound neutralizing antibody response, which indicates much greater potency, durability, and effectiveness against the spike protein mutations. It has also been suggested that the quality of the antibody response improves with increasing exposure to the SARS-CoV-2 antigen [10]. 

Recently, we reported that when patients were spontaneously infected with SARS-CoV-2 after the second or further dose of vaccine, their antibody titers increased to more than 40,000 AU/mL and remained high for several months [11]. 

It was suggested that those with very high antibody titers due to such “hybrid immunity” are expected to have superior protection against the onset and severity of COVID-19, which should make boosters unnecessary for at least six months or more. Participants in our previous study included many employees of the Osaka Dental University Hospital and students of Osaka Dental University, many of whom are healthcare workers at high risk of infection with COVID-19 on a daily basis [11]. In addition, the participants were relatively young people living in metropolitan areas and may not reflect the general population. Elderly people may not be able to produce antibodies due to an insufficient immune response by immune memory B cells, even in the case of a breakthrough infection after two or more doses of vaccine [12]. It is also unclear whether other assay systems for SARS-CoV-2 antibodies would yield similar results. 

In this study, we examined changes in SARS-CoV-2 antibody titers in outpatients who had their SARS-CoV-2 antibody titers measured by self-paid test after vaccination with COVID-19 vaccines in a general internal medicine clinic in a rural area. The study population consisted of patients who attended a private general internal medicine clinic in a rural area of Japan. We consider this population as representative of the older population living in rural areas in Japan. First, we examined the typical antibody titer trends in older people who received their third or fourth booster dose of a COVID-19 vaccine because there have been few papers so far showing such cases in community healthcare institutions. Second, we examined the association between the antibody titers after the second or third dose and the patient’s age or the number of days since vaccination. Third, we examined if breakthrough infections after two or more doses of the COVID-19 vaccine significantly increased the antibody titers.

## 2. Materials and Methods

We conducted a retrospective observational study involving 133 outpatients (49 males and 84 females, median age 70 years) at Ekawa Medical Clinic, Arida City, Wakayama, Japan, between June 2021 to September 2022. The study protocols were approved by the ethics committee of the Osaka Dental University Hospital (2022-10). Written informed consent was obtained from all participants. The titers of antibodies to the receptor-binding domain (RBD) of SARS-CoV-2 were measured in serum samples. 

First, we showed typical antibody titers trends in married couples who were vaccinated with COVID-19 vaccines, after the third and fourth doses.

Second, the correlation between log-transformed antibody titer, age, and the number of days after the second dose of the vaccine was examined in 86 participants (36 males and 50 females) with no history of COVID-19 infection, and the date of vaccination was known. These data were analyzed using histograms and the Shapiro–Wilk test to confirm the normality of the distribution. The same examination was also conducted on 31 subjects (11 males and 20 females) after the third dose of the vaccine. The post-infection antibody titers in participants infected with COVID-19 after two or more doses of the COVID-19 vaccine were also examined.

### 2.1. Serology Assays 

We used the electrochemiluminescence immunoassay system Elecsys Anti-SARS-CoV-2 S (Roche Diagnostics, Mannheim, Germany) to detect IgG antibodies to the receptor-binding domain of the S subunit of the SARS-CoV-2 spike protein, according to the manufacturer’s instructions. The cut-off value set by the manufacturer was 0.8 U/mL. A strong correlation was observed between the measurements of Abbott Architect SARS-CoV-2 IgG II Quant (Abbott Laboratories, Abbott Park, IL, USA), the assay system for the antibody we used in our previous report [11], and those of Elecsys Anti-SARS-CoV-2 S [13].

### 2.2. Statistical Analysis

The results are expressed as mean ± standard deviation (SD). The Pearson’s product-moment correlation coefficient was used to assess the associations between log-transformed SARS-CoV-2 anti-receptor binding domain IgG antibody titers, age, and number of days after vaccination. The normality of the distribution was assessed by the Shapiro–Wilk test, and the transformation to fit a normal distribution was applied to the antibody titers (U/mL) and log-transformed antibody titers. For data analysis, the JMP 13.1 software was used. Statistical significance was set at *p* < 0.05.

## 3. Results

Of the 133 enrolled patients (49 males, 84 females) whose SARS-CoV-2 antibody titers were measured at the Ekawa Medical Clinic, 127 patients (48 males, 79 females) were vaccinated up to the second dose, 97 patients (35 males, 62 females) up to the third dose, 53 patients (21 males, 32 females) up to the fourth dose, and 6 had never been vaccinated. The mean and median age tended to increase with the number of vaccinations. Of the 127 patients vaccinated up to the second dose, 115 patients and 1 patient received the BNT162b2 (BioNTech and Pfizer) and mRNA-1273 (Moderna and Takeda) vaccines, respectively. Of the 97 patients vaccinated up to the third dose, 65 patients and 30 patients received the BNT162b2 and mRNA-1273 vaccines, respectively. Of the 53 vaccinated up to the fourth dose, 39 and 11 patients received the BNT162b2 and mRNA-1273 vaccines, respectively (Table 1, Figure 1).

Figure 2A shows a couple in their 60s who had received the third dose of COVID-19 vaccine. They had both completed the second dose by June 2021, but their antibody titers then decayed, with the wife receiving her third dose in March 2022, and the husband in May 2022. Both of them had much higher antibody titers than after the second dose, up to 10,000 U/mL. The titers then decayed again over the next two to three months, but still remained much higher than the peak after the second dose of vaccine.

Figure 2B shows an example of an elderly couple in their 80s who had received the fourth dose of vaccine. The first two doses of the vaccine were administered by June 2021. The third dose of the vaccine was given in February 2022, and the antibody titer rose to 7000–8000 U/mL immediately. This then dropped over the next 5 months or so, and when the fourth dose of vaccine was given in July 2022, it rose again, but then waned over the next 2 months.

The histograms of the antibody titers and log-transformed antibody titers are shown for 86 participants (36 males, and 50 females) out of the 127 patients who received up to the second dose of vaccine and had antibody titers measured before their third dose and without a history of COVID-19 infection (Figure 3A,B). They had a median age of 68.5 years and were vaccinated with the BNT162b2 vaccine, with a median antibody titer of 1185 U/mL (Table 2). The Shapiro–Wilk test rejected the normality of the antibody titer (U/mL) and the log-transformed antibody titer distributions after the second dose (*p* < 0.001). A negative correlation trend was observed between log-transformed antibody titer and days after the second dose of the vaccine, although this was not significant (*p* = 0.055) (Figure 3C). A significant negative correlation was also observed between log-transformed antibody titer and age (*p* < 0.001) (Figure 3D).

The histograms of the antibody titers and the log-transformed antibody titers are shown for 31 participants out of 97 who had been vaccinated up to the second dose (11 males and 20 females) and had antibody titers measured before the fourth dose without a history of COVID-19 infection (Figure 4A,B). Their median age was 72 years. Of the 31 participants, 27 had received BNT162b2 and 4 had received mRNA-1273. The median antibody titer was 18,300 U/mL, and most likely, it had been boosted from the peak seen after the second dose (Table 2). 

The Shapiro–Wilk test rejected the normality of the antibody titer (U/mL) (*p* = 0.003) and the log-transformed antibody titer distributions after the third dose (*p* = 0.005). No significant correlation was observed between the log-transformed antibody titer after the third dose and the number of post-vaccination days (*p* = 0.414) (Figure 4C). On the other hand, as mentioned above, the log-transformed antibody titer after the third dose showed a significant negative correlation with age (Figure 4D). 

Table 3A shows two cases of spontaneous infection after the second dose of vaccine. A 44-year-old man became infected about six months after the second dose of vaccine, and his antibody titer was over 100,000 U/mL after infection. In a 51-year-old woman, the antibody titer had risen to 26,600 U/mL after infection, but she had received her third vaccination a few days after the antibody titer was measured.

Table 3B shows six cases of spontaneous infection after the third dose of vaccine. Although some of the cases had their antibody titers measured nearly six months after infection, they all maintained high levels of antibody titers. However, despite maintaining high levels of antibody titers, some of them had received the fourth dose of vaccine.

Table 3C shows a case of an elderly woman who was spontaneously infected after the fourth dose of vaccine. This case received the fourth dose of vaccine in July 2022, but she became spontaneously infected in August, and her antibody titer rose to about 120,000 U/mL after infection. Thus, even elderly people seemed to show a marked increase in antibody titer after infection.

## 4. Discussion

The present study showed that SARS-CoV-2 antibody titers were boosted by the third or fourth dose of vaccine, but then declined over time (Figure 2A,B). As observed in this study, immunity acquired by booster vaccination alone is known to fade fast [14]. 

Among the 86 participants whose antibody titers were measured within one month after the second dose of vaccine, age and log-transformed antibody titer showed a significant negative correlation (Figure 3D). A negative correlation trend between the number of days after the second dose of vaccine and the log-transformed antibody titer was observed but was not significant (Figure 3C).

Similarly, in the 31 subjects whose antibody titers were measured within one month after the third dose of vaccine, age and log-transformed antibody titer showed a negative correlation (Figure 4D). It has previously been reported that antibody titers after vaccination tend to decrease with increasing age [15,16,17,18].

The negative correlation between log-transformed antibody titer after the second or third dose of vaccine and age in this study is consistent with these previous reports. The median antibody titer after the third dose of vaccine was more than 10 times higher than that after the second dose of vaccine. In the 1-month post-vaccination follow-up period, no correlation was found between the number of days after the third dose of vaccine and the antibody titer (Figure 4C). It was hypothesized that the third doses of a vaccine would provide stronger immunity and less attenuation of the antibody titers than previous doses of vaccine.

Researchers in Israel studied more than 10,000 healthcare workers who had not previously been infected; all of them received either three or four doses of vaccine. That study showed that up to the second dose of vaccine, the antibody titers tended to drop, but after the third dose of vaccine, the antibody titers were maintained to some extent even after six months [19]. The result that the antibody titers obtained after the third dose of vaccine were less attenuated than those obtained after the second dose of vaccine is consistent with our findings.

On the other hand, in the Israeli study [19], the levels of antibody titers after the third and fourth doses of vaccine did not change much, with the antibody response after the fourth dose of vaccine peaking at about four weeks, declining to the same level as before the fourth dose of vaccine at 13 weeks, and then stabilizing [19]. These results suggest that the fourth dose of vaccine may be less significant in terms of maintenance of the antibody titers. The authors also found that the fourth dose’s clinical efficacy against infection fell rapidly. In fact, after four months, the fourth dose was no better than three doses at preventing infection.

A study derived from the national database in Portugal showed that hybrid immunity, the result of both vaccination and a spontaneous infection by COVID-19, could provide partial protection against reinfection for at least eight months [20]. A survey of healthcare workers in Sweden suggested that those with hybrid immunity had high levels of mucosal antibodies with strong protective effects against infection [21]. It also offered greater than 95% protection against severe disease or hospitalization for between six months and a year after an infection or vaccination, according to estimates from a meta-analysis [22].

In the present study, we showed that the post-infection antibody titers were markedly elevated in patients with post-vaccination breakthrough infection, even in a completely different population and using a different measurement system, from those of our previous study, which also showed similar results. At the same time, there were scattered cases in which even those with sufficiently elevated antibody titers due to breakthrough infection received additional doses of vaccine. It is debatable whether additional vaccinations should be administered when a patient is infected and the antibody titers have increased markedly.

Since the Omicron strain was found to cause low disease severity, the Japanese government has reviewed the status of COVID-19 under the Infectious Disease Control Law and plans to lower it to Category 5, the same as influenza infections, starting in May 2023. In Japan, COVID-19 vaccination has been recommended as frequently as possible in each epidemic surge, but in accordance with this change, there is a proposal to reduce the vaccination frequency to once a year.

Hybrid immunity offers protection against COVID-19 infection and is reported to persist for a relatively long time (between 6–8 months) [23,24]. A systematic review of recent studies has also shown that individuals with hybrid immunity are more protected against the Omicron variant than those with only a history of infection, suggesting that individuals with hybrid immunity may not require a booster dose immediately [22]. In Japan, most cases of SARS-CoV-2 infection occurred after two or more doses of vaccine. In such cases, as shown in our previous report [11], the post-infection antibody titer increased markedly, and the high antibody titer may have continued for a long period of time (more than six months). 

It is necessary to clarify the duration of high antibody titers after breakthrough infection, i.e., whether they last for 6 months to 1 year. In cases where the antibody titers were markedly increased by hybrid immunity from vaccination plus spontaneous infection, the incidence and severity of the COVID-19 were lower, thus providing strong evidence that frequent additional vaccinations are not necessary, at least for recently infected persons, and that annual vaccinations are sufficient. 

This study has several limitations. First, the size of the population included in the study was small. Second, there was no information on the SARS-CoV-2 variants. However, these limitations are difficult to overcome because the study was conducted at a single private clinic. Third, the natural immunity to SARS-CoV-2 after vaccination has not been studied yet. In addition to the induction of antibodies, the effect of natural immunity is considered significant regarding hybrid immunity. As for the natural immunity after COVID-19 infection, the effects of CD4+ T cells, CD8+ T cells, and memory B cells have been shown to be maintained for more than 8 months [25,26]. Studies in Qatar have also shown that the natural immune protection against the SARS-CoV-2 infection wanes over time, but the prophylactic effect against the severe disease forms of COVID-19 remains strong [27].

In conclusion, as we mentioned in a previous report [11], our study found that an increase in the antibody titers against SARS-CoV-2 is an important indicator of the effect of COVID-19 vaccines. In addition, there were cases such as breakthrough infections after two or three doses of the COVID-19 vaccine with markedly increased antibody titers in participants who still received further booster doses of the mRNA vaccine. In addition, this study found that the antibody titers after the third vaccination did not attenuate after a short follow-up period of one month, while they tended to attenuate after the second vaccination. In other words, three vaccinations are thought to produce stronger immunity. Although it is desirable to measure the antibody titers and prioritize those people with low antibody titers for booster vaccination, rather than blindly recommending booster vaccination to the entire population, it may be difficult to see how the financial, personnel, time, and educational costs can be afforded. As this was a small retrospective study, larger-scale longitudinal studies of antibody titers are warranted.

## Figures and Tables

**Figure 1 medicines-10-00027-f001:**
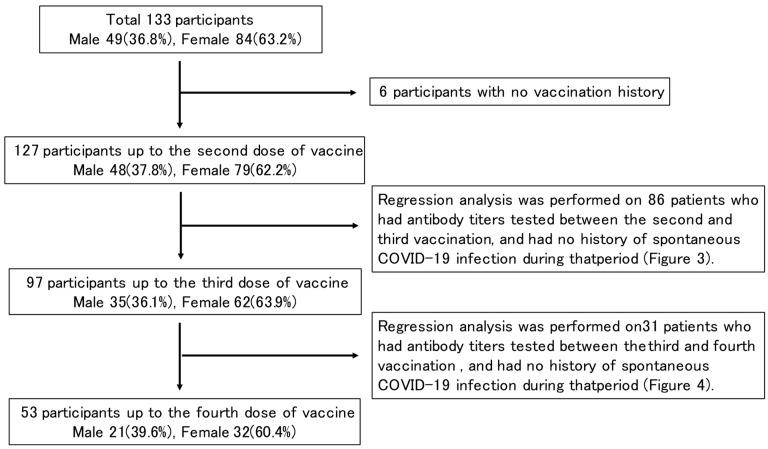
Study design flowchart.

**Figure 2 medicines-10-00027-f002:**
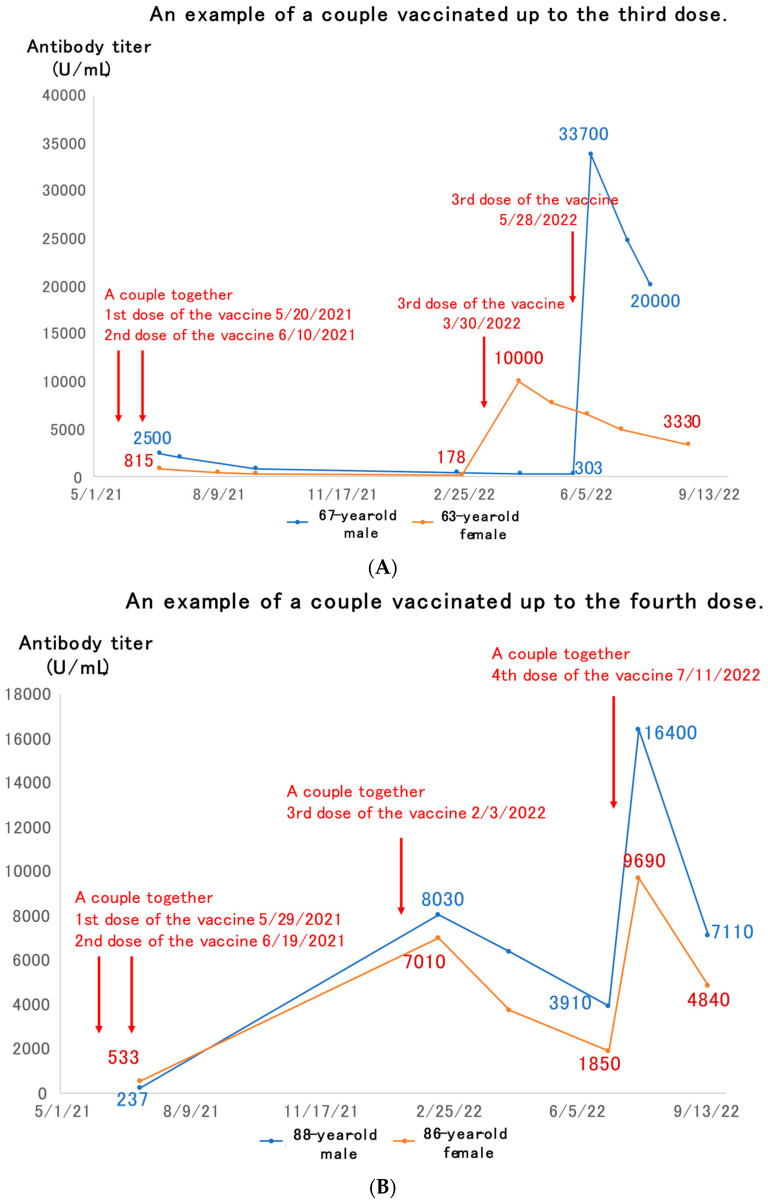
Examples of couples vaccinated up to the third (**A**) or fourth (**B**) dose of vaccine.

**Figure 3 medicines-10-00027-f003:**
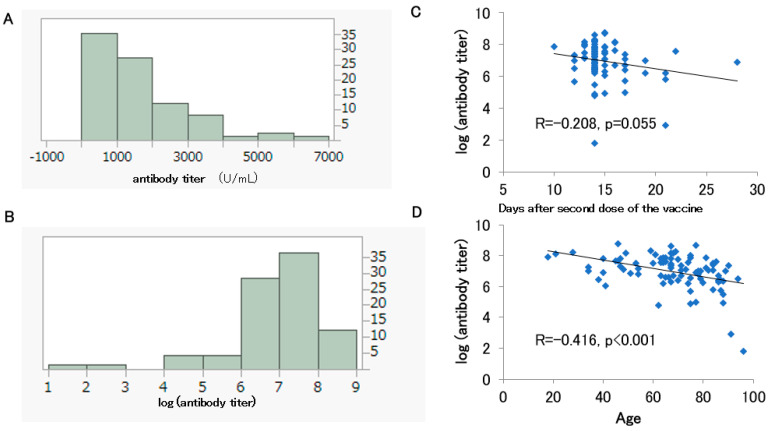
Histograms of the antibody titers (**A**) and log-transformed antibody titers (**B**) for 86 participants who had been vaccinated up to the second dose. Correlation between the days after the second dose and the log-transformed antibody titer (**C**). Correlation between the participants’ age and the log-transformed antibody titer (**D**).

**Figure 4 medicines-10-00027-f004:**
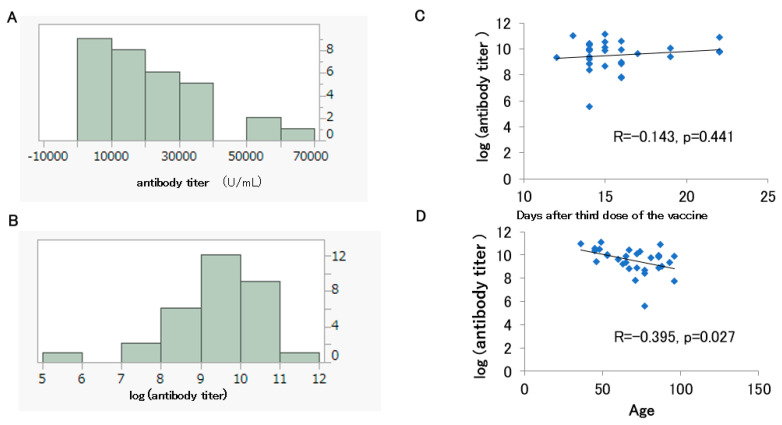
Histograms of the antibody titers (**A**) and log-transformed antibody titers (**B**) for 31 participants who had been vaccinated up to the third dose. Correlation between the days after the third dose and the log-transformed antibody titer (**C**). Correlation between the participant’s age and the log-transformed antibody titer (**D**).

**Table 1 medicines-10-00027-t001:** Participants’ characteristics up to the second, third, and fourth vaccine doses.

	Total Participants	Participants up to the Second Dose	Participants up to the Third Dose	Participants up to the Fourth Dose
**Total**	133	127	97	53
**Male, n (%)**	49 (36.8)	48 (37.8)	35 (36.1)	21 (39.6)
**Female, n (%)**	84 (63.2)	79 (62.2)	62 (63.9)	32 (60.4)
**Mean age** **Median age**	65.7 ± 19.1 70	66.4 ± 18.7 70	70.2 ± 16.1 72	74.3 ± 12.5 75
**Type of vaccine**	**BNT162b2 (BioNTech and Pfizer), n (%)**	115 (90.5)	65 (67.0)	39 (73.6)
	**mRNA-1273 (Moderna and Takeda), n (%)**	1 (0.8)	30 (30.9)	11 (20.7)
	**Unknown, n (%)**	11 (8.7)	2 (2.1)	3 (5.7)
**No vaccination history, n (%)**	6 (4.5)			

**Table 2 medicines-10-00027-t002:** Analyzed participants up to the second and third vaccine dose.

		Analyzed Participants up to the Second Dose	Analyzed Participants up to the Third Dose
**Total**		86	31
**Male, n (%)**		36 (41.9)	11 (35.5)
**Female, n (%)**		50 (58.1)	20 (64.5)
**Mean age**		67.3 ± 17.2	69.9 ± 16.8
**Median age**		68.5	72
**Type of vaccine**	**BNT162b2 (BioNTech and Pfizer), n (%)**	86 (100)	27 (87.1)
	**mRNA-1273 (Moderna and Takeda), n (%)**	0 (0)	4 (12.9)
**Median antibody titer (U/mL)**		1185	18,300
**Median log (antibody titer)**		7.1	9.8
**Mean number of days between the vaccination and the measurement day (Mean ± standard deviation)**		14.9 ± 2.5	15.6 ± 2.6

**Table 3 medicines-10-00027-t003:** Participants who became infected after the second (A), third (B), and fourth (C) dose of vaccine.

**A. Participants Who Became Infected after the Second Dose of Vaccine.**							
**Case**	**1st Vaccination Date**	**Date of Infection**	**Days from 2nd Vaccination to Infection**	**Post-Infection Antibody Test Date**	**Post-Infection Antibody Titer (U/mL)**	**Number of Days from Infection to Antibody Titer Measurement**	**3rd Vaccination Date**
	**2nd Vaccination Date**						
**44M**	11 August 2021	3 March 2022	183	9 April 2022	113,000	37	
	1 September 2021						
**51F**	unknown	15 January 2022		4 March 2022	26,600	48	9 March 2022
	unknown						
**B. Participants who became infected after the third dose of vaccine**							
**Case**	**1st Vaccination Date**	**Date of Infection**	**Days from 3rd Vaccination to Infection**	**Post-Infection Antibody Test Date**	**Post-Infection Antibody Titer (U/mL)**	**Number of Days from Infection to Antibody Titer Measurement**	**4th Vaccination Date**
	**2nd Vaccination Date**						
	**3rd Vaccination Date**						
**65F**	22 June 2021	29 March 2022	36	12 April 2022	20,500	14	3 August 2022
	13 July 2021						
	21 February 2022						
**75F**	23 June 2021	14 July 2022	118	25 July 2022	12,200	11	
	14 July 2021						
	18 March 2022						
**57F**	unknown	18 August 2022	160	13 September 2022	59,600	26	
	unknown						
	11 March 2022						
**75M**	26 June 2021	8 August 2022	156	26 August 2022	41,000	18	26 August 2022
	17 July 2021						
	5 March 2022						
**71F**	10 July 2021	10 August 2022	153	26 August 2022	9840	16	26 August 2022
	31 July 2021						
	10 March 2022						
**71F**	unknown	11 March 2022	31	25 April 2022	63,400	45	3 August 2022
	unknown						
	8 February 2022						
**C. Participants who became infected after the fourth dose of vaccine**							
**Case**	**1st Vaccination Date**	**Date of Infection**	**Days from 4th Vaccination to Infection**	**Post-Infection Antibody Test Date**	**Post-Infection Antibody Titer (U/mL)**	**Number of Days from Infection to Antibody Titer Measurement**	
	**2nd Vaccination Date**						
	**3rd Vaccination Date**						
	**4th Vaccination Date**						
**85F**	unknown	21 August 2022	44	13 September 2022	122,000	23	
	unknown						
	8 February 2022						
	8 July 2022						

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
