# Peer review of "Trends in Antibody Titers after SARS-CoV-2 Vaccination—Insights from Self-Paid Tests at a General Internal Medicine Clinic"

_medicines, 2023, doi:10.3390/medicines10040027_

Round 1
Reviewer 1 Report
The paper is very interesting, but i have some questions to the authors :
1-Table 1A shows two cases of spontaneous infection after the second dose of vaccine. A 44-year-old man became infected about six months after the second dose of vaccine, and his antibody titer was over 100,000U/mL after infection. In a 51-year-old woman, the antibody titer had risen to 26,600U/mL after infection, but she had received her third vaccination a few days after the antibody titer was measured.
What if the age was changed? Does this affect the results?
Table 1B shows six cases of spontaneous infection after the third dose of vaccine. Although some of the cases had their antibody titers measured nearly six months after infection, they all maintained high levels of antibody titers. However, despite maintaining high levels of antibody titers, some of them had received the fourth dose of vaccine
What if it was the third dose? Does this affect the results, and how?
In Table one, why do authors use these ages specifically?
Is it random, or what?
In the Statistical analysis section, what is the main advantage of the software sued over Minitab and SPSS?
The Statistical significance was set at p < 0.05. What would happen if this value changed?
The main differences should be mentioned between this study and the old ones.
The novelty of this work should be cleared
The paper is interesting, and the study is good
Author Response
Reviewer 1
The paper is very interesting, but i have some questions to the authors :
1-Table 1A shows two cases of spontaneous infection after the second dose of vaccine. A 44-year-old man became infected about six months after the second dose of vaccine, and his antibody titer was over 100,000U/mL after infection. In a 51-year-old woman, the antibody titer had risen to 26,600U/mL after infection, but she had received her third vaccination a few days after the antibody titer was measured. What if the age was changed? Does this affect the results?
We would like to thank the reviewer for his constructive criticism and helpful suggestions, which helped us revise and improve our manuscript. We would also like to thank the reviewer for his valuable remarks.
It is possible that older age may result in lower antibody titers after infection. We are presenting these two cases because we happen to have a 44-year-old man and a 51-year-old woman in our database who were spontaneously infected after the second dose of vaccine.
Table 1B shows six cases of spontaneous infection after the third dose of vaccine. Although some of the cases had their antibody titers measured nearly six months after infection, they all maintained high levels of antibody titers. However, despite maintaining high levels of antibody titers, some of them had received the fourth dose of vaccine.
What if it was the third dose? Does this affect the results, and how?
In Table one, why do authors use these ages specifically?
Is it random, or what?
In the Statistical analysis section, what is the main advantage of the software used over Minitab and SPSS?
The Statistical significance was set at p < 0.05. What would happen if this value changed?
The main differences should be mentioned between this study and the old ones.
The novelty of this work should be cleared
The paper is interesting, and the study is good
Whether it is the third or fourth vaccination, the antibody titer is expected to increase further when a booster vaccination is given to those with antibody titers in the tens of thousands. Although it is unclear whether a marked increase in antibody titer will have any adverse effects on the body, mRNA vaccination is associated with a high frequency of adverse reactions such as fever, and vaccination of persons with sufficient antibody titer should be discouraged.
JMP 13.1 software was used for statistical analysis. This software is not particularly advantageous compared to SPSS, but it is widely used.
In our previous study, including our previous study, antibody titers tended to decline over time after vaccination, both after the second and third vaccination. In our study, we found that antibody titers after the third vaccination did not attenuate after a short follow-up period of one month, while antibody titers after the second vaccination tended to attenuate. In other words, third vaccinations are thought to produce stronger immunity.
In our previous study, we also observed that antibody titers increased to more than 10000AU/mL in breakthrough infection after vaccination and remained high for more than six months. In this study, we found that even though antibody titers were markedly elevated after breakthrough infection, there were scattered cases of patients receiving further booster doses of mRNA vaccine. We believe that these are the novelties of this study.

Reviewer 2 Report
Medicine-2278461: Trends in Antibody Titers after SARS-CoV-2 Vaccination
Insights from a Self-Paid Tests at a General Internal Medicine Clinic
I would thank the authors for this interesting idea (manuscript). However I found that the manuscripts has presented multiple that make it of poor quality
A lack of scientific rigor is apparent in all parts of the manuscript (abstract, introduction, results, discussion...) and in all its characteristics (writing, summarizing, language, analysis) that are necessary for the manuscript to be published.
My main concern is related to the objectives of the study. Is this manuscript an article? a brief report or a case report?. In fact, the authors tend to describes some the results of two (or more) individuals and try to generalize them to all the study population and discuss them. Thus, you should precise their methodology and choose the adequate type of the paper.
The other concern is related to the writing. The authors used long paragraphs (in the introduction and the discussion) and sometimes without any reference (see below).
Thus, I suggest to the authors (by the permission of the editor) to take more time and revise and rewrite all the parts of the manuscript before its resubmission.
I can tell you also some comments and suggestion:
Title:
Delete “a” (self-paid tests) from the title
Abstract:
1. Delete the discussion from the abstract
Introduction
You need to add more reference in the introduction and define your hypothesis. Two references for an introduction are very little. Using one reference for 5 paragraphs is not sufficient. Please revise
Material and Methods:
Correct the first paragraph please: retrospective, observational, longitudinal.. Try to be more concise please.
Delete respectively from the end of the second paragraph
Third paragraph: “... and the date of vaccination was known”. try to reformulate and delete the repetition please.
Results:
I suggest revising all your results according to your choice for the type of the manuscript.
Also, you should begin with the descriptive features of the study population.
Additionally the separation between the results of each vaccine dose is not clear. Try to reorganize please
“since all cases.....about two weeks after vaccination”. Delete this paragraph.
Discussion:
The discussion should be updated and adapted according to the type of the paper. As explained before we can not associate between some cases and the total population and discuss them together. You should discuss each result alone.
Also, the discussion lacks references. Paragraphs 3 and 4 are without any reference
At last, all the above are “some of the remarks” that I found. I recommend to the authors to revise substantially all the manuscript.

Author Response
Reviewer 2
I would thank the authors for this interesting idea (manuscript). However I found that the manuscripts has presented multiple that make it of poor quality
A lack of scientific rigor is apparent in all parts of the manuscript (abstract, introduction, results, discussion...) and in all its characteristics (writing, summarizing, language, analysis) that are necessary for the manuscript to be published.
My main concern is related to the objectives of the study. Is this manuscript an article? a brief report or a case report?. In fact, the authors tend to describes some the results of two (or more) individuals and try to generalize them to all the study population and discuss them. Thus, you should precise their methodology and choose the adequate type of the paper.
The other concern is related to the writing. The authors used long paragraphs (in the introduction and the discussion) and sometimes without any reference (see below).
Thus, I suggest to the authors (by the permission of the editor) to take more time and revise and rewrite all the parts of the manuscript before its resubmission.
I can tell you also some comments and suggestion:
We would like to thank the reviewer for his constructive criticism and helpful suggestions, which helped us revise and improve our manuscript. We would also like to thank the reviewer for his valuable remarks.
This study is a report on a small population in which antibody titers were measured after COVID-19 vaccination in a private clinic. Therefore, we can only present a few cases of COVID-19 in this target population. We would like to request considering whether there are similar cases in larger populations in the future, but this is beyond our ability. Therefore, although this study is an article based on case reports, we would like to see future studies to see if it can be generalized to larger populations.
Title:
Delete “a” (self-paid tests) from the title
Abstract:
- Delete the discussion from the abstract
Introduction
You need to add more reference in the introduction and define your hypothesis. Two references for an introduction are very little. Using one reference for 5 paragraphs is not sufficient. Please revise
Material and Methods:
Correct the first paragraph please: retrospective, observational, longitudinal.. Try to be more concise please.
Delete respectively from the end of the second paragrap
Third paragraph: “... and the date of vaccination was known”. try to reformulate and delete the repetition please.
Results:
I suggest revising all your results according to your choice for the type of the manuscript.
Also, you should begin with the descriptive features of the study population.
Additionally the separation between the results of each vaccine dose is not clear. Try to reorganize please
“since all cases.....about two weeks after vaccination”. Delete this paragraph.
Discussion:
The discussion should be updated and adapted according to the type of the paper. As explained before we can not associate between some cases and the total population and discuss them together. You should discuss each result alone.
Also, the discussion lacks references. Paragraphs 3 and 4 are without any reference
At last, all the above are “some of the remarks” that I found. I recommend to the authors to revise substantially all the manuscript.
We have removed the points you indicated should be deleted.
We have added the following sentences to the introduction.
“The purpose of this study was to demonstrate the fact that "breakthrough infection after 2 or 3 doses of COVID-19 vaccine markedly increases antibody titers" in a population independent of our previous study.”
“The study population consists of patients attending a private internal medicine clinic in a rural area of Japan. We consider this to be an average population of elderly people living in rural areas in Japan.”
We have added references to the introduction, so please check.

Reviewer 3 Report
It is necessary to reanalyze the data with the consultation of people in the field of biostatistics and serology.
- the first sentence in the abstract and the same in the introduction
- "The rise in antibody titers against the novel coronavirus (SARS-CoV-2), and its duration is considered an important indicator for confirming the effectiveness of the COVID-19 vaccine, and self -paid tests of antibody titer are conducted at many facilities nationwide.” The increase in the titre of antibodies and their persistence is not an indicator confirming the effectiveness of the vaccines, but only determines the immunogenicity of the product and the kinetics of its effect after application. These are parameters that determine the activity of the product at the laboratory level. Low levels of antibodies with high affinity and avidity to the antigen, although of shorter duration, will be more effective. Thus, immunogenicity and T1/2 of persistence of specific antibodies may or may not be an indicator of vaccine effectiveness in terms of the desired effect. Efficiency is the result of many different factors. These parameters are predictive only. Please clarify this sentence.
- The same statement in another sentence in the introduction
- "The strength of "hybrid immunity" with the rise in, and duration of, antibody titers against SARS-CoV-2 after post-vaccination infection are considered important indicators for confirming the efficacy of COVID-19 vaccines” Please change.
- The conclusion that people with low antibody titers should be vaccinated is obvious and very general. The work has not been shown to bring anything new to the area;
- no vaccines were administered to the test subjects
- „Recently we had reported that when patients are spontaneously infected with SARSCoV- 2 after the second or more dose of vaccine, antibody titers increased to more than 40,000 AU/mL and remained high for several months [2].” This sentence brings nothing without comparison to the results of the control and showing the difference. - It is inappropriate to include a description of a previous study in the introduction. It starts with the sentence – “Participants in our previous study included many employees of Osaka Dental University Hospital and students of Osaka Dental University, many of whom are healthcare workers at high risk of infection with COVID-19 on a daily basis.” The characteristics of the study population are given in the materials and methods and apply to the current study, not the previous one.
- In the introduction, the authors state that "In this study, we examined changes in SARS-CoV-2 antibody titers in outpatients who had their SARS-CoV-2 antibody titers measured by self-paid test after vaccination with the COVID-19 vaccine in general internal medicine clinics in rural areas.” And in the first sentence of the materials and methods - "This was a single-center retrospective study." What's the truth ?
- „A strong correlation was observed between the measurements of Abbott Architect SARSCoV-2 IgG II Quant (Abbott Laboratories, Illinois, USA), the assay system for the antibody we used in our previous report [2], and those of Elecsys Anti-SARS-CoV-2 S [3].” Strong means what kind and what does it concern?
- Statistical analysis - no statistical inference path was given, wrong tests were used. The distribution of antibody titers is log-linear and the arithmetic mean and parametric tests based thereon were used as for the normal distribution. Re-statistical correct processing of the results is required.
- providing titer values is correct, but it is also important to provide change parameters, e.g. multiplicity or %.
- “Since all cases were tested at private clinics at their own expense, most people came to see if they had enough antibody titers after vaccination, so antibody titers were measured within one month after vaccination. On an average, antibody titers were measured about two weeks after vaccination.” This assertion is incomprehensible.
- it is not specified what measures of central tendency are given, what for antibody titer, what for age, etc. - Figure 2 and 3 - 2B and 3B are correct, 2A and 3A are wrong.
The material and idea is to be published, but on the condition of a thorough re-analysis of the results with consultation of a biomedical statistician, correction of the introduction and discussion, supplementation of materials and methods and a different approach to the conclusions resulting from the results. It is also necessary to demonstrate the minimum novelty of the work.
Author Response
Reviewer 3
It is necessary to reanalyze the data with the consultation of people in the field of biostatistics and serology.
- the first sentence in the abstract and the same in the introduction
- "The rise in antibody titers against the novel coronavirus (SARS-CoV-2), and its duration is considered an important indicator for confirming the effectiveness of the COVID-19 vaccine, and self -paid tests of antibody titer are conducted at many facilities nationwide.” The increase in the titre of antibodies and their persistence is not an indicator confirming the effectiveness of the vaccines, but only determines the immunogenicity of the product and the kinetics of its effect after application. These are parameters that determine the activity of the product at the laboratory level. Low levels of antibodies with high affinity and avidity to the antigen, although of shorter duration, will be more effective. Thus, immunogenicity and T1/2 of persistence of specific antibodies may or may not be an indicator of vaccine effectiveness in terms of the desired effect. Efficiency is the result of many different factors. These parameters are predictive only. Please clarify this sentence.
We would like to thank the reviewer for his constructive criticism and helpful suggestions, which helped us revise and improve our manuscript. We would also like to thank the reviewer for his valuable remarks.
As you have pointed out, the rise in antibody titer and its persistence is not an indicator to confirm the efficacy of the vaccine, but rather to determine the immunogenicity of the vaccine and the dynamics of its efficacy after application, and does not inherently reflect clinical efficacy. However, especially in Japan, the efficacy of COVID-19 vaccine tends to be evaluated by the rise in antibody titer. This can be called "supremacy of antibody titer," so to speak, and we believe that this needs to be reviewed in the future.
We have added the following sentences in the introduction.
“Originally, the rise in antibody titer and its persistence is not an indicator to confirm the efficacy of the vaccine, but rather to determine the immunogenicity of the vaccine and the dynamics of its efficacy after application, and does not inherently reflect clinical efficacy. However, especially in Japan, the efficacy of COVID-19 vaccine tends to be evaluated by the rise in antibody titer.”
- The same statement in another sentence in the introduction
- "The strength of "hybrid immunity" with the rise in, and duration of, antibody titers against SARS-CoV-2 after post-vaccination infection are considered important indicators for confirming the efficacy of COVID-19 vaccines” Please change.
- The conclusion that people with low antibody titers should be vaccinated is obvious and very general. The work has not been shown to bring anything new to the area;
- no vaccines were administered to the test subjects
- „Recently we had reported that when patients are spontaneously infected with SARSCoV- 2 after the second or more dose of vaccine, antibody titers increased to more than 40,000 AU/mL and remained high for several months [2].” This sentence brings nothing without comparison to the results of the control and showing the difference. - It is inappropriate to include a description of a previous study in the introduction. It starts with the sentence – “Participants in our previous study included many employees of Osaka Dental University Hospital and students of Osaka Dental University, many of whom are healthcare workers at high risk of infection with COVID-19 on a daily basis.” The characteristics of the study population are given in the materials and methods and apply to the current study, not the previous one.
We have corrected the part of your comment about repeated expressions.
We have added the following sentences.
“The study population consists of patients attending a private internal medicine clinic in a rural area of Japan. We consider this to be an average population of elderly people living in rural areas in Japan.”
We believe the novelty of this study is that we found that there were scattered cases of breakthrough infection after two or three doses of COVID-19 vaccine and markedly increased antibody titers, but still received further booster doses of mRNA vaccine. In addition, this study found that antibody titers after the third vaccination did not attenuate after a short follow-up period of one month, while antibody titers after the second vaccination tended to attenuate. In other words, three vaccinations are thought to produce stronger immunity.
These are the novelties of this study, and we have added them to the discussion section.
- In the introduction, the authors state that "In this study, we examined changes in SARS-CoV-2 antibody titers in outpatients who had their SARS-CoV-2 antibody titers measured by self-paid test after vaccination with the COVID-19 vaccine in general internal medicine clinics in rural areas.” And in the first sentence of the materials and methods - "This was a single-center retrospective study." What's the truth ?
This study is a retrospective study of the evolution of antibody titers in patients with self-tested SARS-CoV-2 antibody titers after COVID-19 vaccination in a rural area.
The above is described in the MATERIALS AND METHODS section.
- „A strong correlation was observed between the measurements of Abbott Architect SARSCoV-2 IgG II Quant (Abbott Laboratories, Illinois, USA), the assay system for the antibody we used in our previous report [2], and those of Elecsys Anti-SARS-CoV-2 S [3].” Strong means what kind and what does it concern?
In their study, the correlation between the anti-RBD antibody titer measured by the Architect assay and the antibody titer measured by the Elecsys assay at S1 protein are shown in scatter plots with the regression lines. Spearman’s correlation coefficients were 0.74 (P < 0.01) for S1.
- Statistical analysis - no statistical inference path was given, wrong tests were used. The distribution of antibody titers is log-linear and the arithmetic mean and parametric tests based thereon were used as for the normal distribution. Re-statistical correct processing of the results is required.
- providing titer values is correct, but it is also important to provide change parameters, e.g. multiplicity or %.
- it is not specified what measures of central tendency are given, what for antibody titer, what for age, etc. - Figure 2 and 3 - 2B and 3B are correct, 2A and 3A are wrong.
As you indicated, the distribution of antibody titers is log-linear, and we believe that log-transformed antibody titers should be shown in the scatterplots. Therefore, Figures 2A and 3A should be omitted and Figures 2B and 3B should remain.
- “Since all cases were tested at private clinics at their own expense, most people came to see if they had enough antibody titers after vaccination, so antibody titers were measured within one month after vaccination. On an average, antibody titers were measured about two weeks after vaccination.” This assertion is incomprehensible.
We used the above expression because patients who want to check whether antibody titers have increased after vaccination tend to come for antibody titer measurement immediately after vaccination (within one month). However, since it seems to be difficult to understand, we will remove the relevant section.

Round 2
Reviewer 2 Report
I would like to thank the authors for their efforts to improve the quality of the manuscript. However, the main concern about the type of the study without response and the author's justification did not convince me. Also, the lack of the descriptive characteristics of the population make of the manuscript of very limited importance.
Additionnally, a lack of scientific rigours still be apparent. Deleting the term "discussion" from the abstract and associate the paragraph of the discussion with the conclusion is inacceptable and shows the best example.
Also, almost all added paragraph are self statements paragraphs and without references.
On the other hand, the authors just add some references to their paragraphs without any efforts to reformulate. The manuscript needs to be enriched with references in relation with subject.
All figures have tow titles.
Author Response
Reviewer 2
I would like to thank the authors for their efforts to improve the quality of the manuscript. However, the main concern about the type of the study without response and the author's justification did not convince me. Also, the lack of the descriptive characteristics of the population make of the manuscript of very limited importance.
Additionnally, a lack of scientific rigours still be apparent. Deleting the term "discussion" from the abstract and associate the paragraph of the discussion with the conclusion is inacceptable and shows the best example.
Also, almost all added paragraph are self statements paragraphs and without references.
On the other hand, the authors just add some references to their paragraphs without any efforts to reformulate. The manuscript needs to be enriched with references in relation with subject.
All figures have tow titles.
Thank you for your additional comments.
・Regarding the "descriptive characteristics of the population," Table 1 summarizes the overall population, and Table 2 summarizes the population used for regression analysis.
・For visual clarity, the study design is summarized in Figure 1.
・References are listed in the introduction and discussion sections, with articles mainly on hybrid immunity, which is the main subject of this paper.
・In the Conclusion section, the contentious issue of additional vaccine doses for those with hybrid immunity was further emphasized.
・About the issue of the study type, as we mentioned in the previous peer review, we consider it to be an original article though based on case reports. Therefore, our view is that this is an original article.

Reviewer 3 Report
Only some comments have been taken into account.
The added sentence in the introduction is not correct:
"Originally, the rise in antibody titer and its persistence is not an indicator to confirm the efficacy of the vaccine, but rather to determine the immunogenicity of the vaccine and the dynamics of its efficacy after application, and does not inherently reflect clinical efficacy. However, especially in Japan, the efficacy of COVID-19 vaccine tends to be evaluated by the rise in antibody titer."
The reviewer's intent was to point out to the authors that they did not distinguish between laboratory and clinical efficacy of the vaccine, which they continue to affirm. The term "efficacy" should not be used without specifying what efficacy is meant.
The statement that vaccine efficacy is referred to in this way in Japan is not true. Without explicit specification, the term "efficacy" is entitled to clinical effectiveness. All major registration agencies require data on, among other things, the pharmacokinetics of the induced effect in the form of antibodies, the determination of their specificity and the clinical efficacy of the vaccine.
The authors state well that the induced antibody level indicates the immunogenicity of the vaccine which indirectly and predictively should correlate with the clinical efficacy of the product provided its specificity is maintained, but it is the efficacy in inducing a response.
The type of vaccine(s) with which the subjects were vaccinated is still not specified.
2.2 Statistical analysis "The results are expressed as mean ± standard deviation (SD). Pearson's product-moment correlation coefficient was used to assess the associations (...)"
The response states that Spearman's correlation coefficients were used. What is the truth?
The arithmetic mean was used, which is inappropriate as a measure of central tendency for log-normal distributions ?
Mean age should not be presented as arithmetic mean only median a for antibody levels.
The statistical inference path is still not provided. The distribution of antibody titers is log-linear and the arithmetic mean was used and parametric tests based on it as for a normal distribution.
It was not checked whether the log-transformed antibody titer had a normal distribution.
Author Response
Reviewer 3
Only some comments have been taken into account.
The added sentence in the introduction is not correct:
"Originally, the rise in antibody titer and its persistence is not an indicator to confirm the efficacy of the vaccine, but rather to determine the immunogenicity of the vaccine and the dynamics of its efficacy after application, and does not inherently reflect clinical efficacy. However, especially in Japan, the efficacy of COVID-19 vaccine tends to be evaluated by the rise in antibody titer."
The reviewer's intent was to point out to the authors that they did not distinguish between laboratory and clinical efficacy of the vaccine, which they continue to affirm. The term "efficacy" should not be used without specifying what efficacy is meant.
The statement that vaccine efficacy is referred to in this way in Japan is not true. Without explicit specification, the term "efficacy" is entitled to clinical effectiveness. All major registration agencies require data on, among other things, the pharmacokinetics of the induced effect in the form of antibodies, the determination of their specificity and the clinical efficacy of the vaccine.
The authors state well that the induced antibody level indicates the immunogenicity of the vaccine which indirectly and predictively should correlate with the clinical efficacy of the product provided its specificity is maintained, but it is the efficacy in inducing a response.
Thank you for the additional comments.
Surely the term "efficacy" was used to refer to the vaccine pharmacokinetics, i.e., the increase in antibody titer, and it is still unclear whether it can be used to describe clinical benefits, such as reduced incidence or severity of COVID-19. Therefore, the term "effect" denotes the increase in the antibody titer or the vaccine’s effects, while "clinical efficacy" refers to the clinical benefit.
The type of vaccine(s) with which the subjects were vaccinated is still not specified.
Table 1 and Table 2 have detailed information about the types of vaccines used.
2.2 Statistical analysis "The results are expressed as mean ± standard deviation (SD). Pearson's product-moment correlation coefficient was used to assess the associations (...)"
The response states that Spearman's correlation coefficients were used. What is the truth?
We used Pearson's correlation coefficient in this article, while Spearman's correlation coefficient was used in the cited reference (12. Matsuura T, Fukushima W, Nakagama Y, et al. Vaccine. Sep 09 2022;40(38):5631-5640). This paper showed a strong correlation between the values of the SARS-CoV-2 IgG II Quant assay, which was the kit used in our previous study, and those of the Elecsys Anti-SARS-CoV-2 S assay kit used in this study, which employed the Spearman's correlation coefficient. In the previous peer review, when we were asked, "How is a strong correlation indicated?" our reply was, "It is shown by Spearman's correlation coefficient.” Therefore, we apologize for the confusion.
The arithmetic mean was used, which is inappropriate as a measure of central tendency for log-normal distributions?
Mean age should not be presented as arithmetic mean only median a for antibody levels.
The statistical inference path is still not provided. The distribution of antibody titers is log-linear and the arithmetic mean was used and parametric tests based on it as for a normal distribution.
It was not checked whether the log-transformed antibody titer had a normal distribution.
We showed the histograms of the antibody titers distribution and log-transformed antibody titers in Figure 3A, B, and Figure 4A, B. As it was pointed out, the distribution of antibody titer was log-linear (Figure 3A, 4A). Although the logarithmic transformation makes the distribution closer to a normal type, the normality of the Shapiro-Wilk test could not be confirmed with this number of samples. The median age was also shown (Table 1, 2), as well as the median and log-transformed antibody titers (Table 2).
